# A Dynamic Compartmental Model to Explore the Optimal Strategy of Varicella Vaccination: An Epidemiological Study in Jiangsu Province, China

**DOI:** 10.3390/tropicalmed8010017

**Published:** 2022-12-27

**Authors:** Xiang Sun, Chenxi Dai, Kai Wang, Yuanbao Liu, Xinye Jin, Congyue Wang, Yi Yin, Zhongxing Ding, Zhenzhen Lu, Weiming Wang, Zhiguo Wang, Fenyang Tang, Kaifa Wang, Zhihang Peng

**Affiliations:** 1Department of Expanded Program on Immunization, Jiangsu Provincial Center for Disease Control and Prevention, Nanjing 210009, China; 2Department of Biomedical Engineering and Imaging Medicine, Army Medical University, Chongqing 400038, China; 3The First Clinical Medical College of Nanjing Medical University, Nanjing 210029, China; 4Department of Epidemiology and Biostatistics, School of Public Health, Nanjing Medical University, Nanjing 211166, China; 5Department of Mathematics and Statistics, Huaiyin Normal University, Huaian 223300, China; 6Department of Mathematics and Statistics, Southwest University, Chongqing 400715, China

**Keywords:** varicella immunization, two-dose varicella vaccination, age-structured dynamic model, basic reproduction number

## Abstract

Varicella (chickenpox) is highly contagious among children and frequently breaks out in schools. In this study, we developed a dynamic compartment model to explore the optimal schedule for varicella vaccination in Jiangsu Province, China. A susceptible-infected-recovered (SIR) model was proposed to simulate the transmission of varicella in different age groups. The basic reproduction number was computed by the kinetic model, and the impact of three prevention factors was assessed through the global sensitivity analysis. Finally, the effect of various vaccination scenarios was qualitatively evaluated by numerical simulation. The estimated basic reproduction number was 1.831 ± 0.078, and the greatest contributor was the 5–10 year-old group (0.747 ± 0.042, 40.80%). Sensitivity analysis indicated that there was a strong negative correlation between the second dose vaccination coverage rate and basic reproduction number. In addition, we qualitatively found that the incidence would significantly decrease as the second dose vaccine coverage expands. The results suggest that two-dose varicella vaccination should be mandatory, and the optimal age of second dose vaccination is the 5–10 year-old group. Optimal vaccination time, wide vaccine coverage along with other measures, could enhance the effectiveness of prevention and control of varicella in China.

## 1. Background

Varicella is a highly contagious disease caused by varicella-zoster virus (VZV) [1,2]. VZV infection arises with the inhalation of aerosols containing pathogens or direct body contact. In the absence of varicella vaccination, varicella affects almost everyone by mid-adulthood [3]. The average annual incidence of varicella increased by 30% from 2005 to 2015 [4], and the number of people with varicella in China continues to rise. A seroepidemiological study in Suzhou, China, showed that about half of Chinese children aged 4–6 years were at high risk of contracting chickenpox virus [5]. In China, the direct and indirect damage, burden of disease and cost of medical services due to varicella is becoming increasingly severe [6,7,8]. All these pose a high economic burden on the public health system [9,10].

Universal varicella vaccination (UVV) has proven to be the most effective way to control the spread of VZV [1,2]. In the United States, every child is required to receive a one-dose varicella vaccination (VarV) after 1995 (the dose doubled in 2006), causing a dramatic decrease of the morbidity, hospitalization rate and mortality [2,11]. This immunization strategy has also been implemented in Germany, Australia, and Canada [12]. However, despite that varicella vaccine is recommended by the World Health Organization (WHO) [13], some developing countries have not incorporated it in national programs [14,15]. At present, some provinces and cities in China have started to implement the two-dose vaccination strategy, such as Beijing, Tianjin, Shanghai and Guangdong, which also use the above vaccination strategy.

In Jiangsu Province, the number of varicella cases has increased every year, even exponentially after 2014. Varicella emergency frequently occurs, with 128 outbreaks in 2017 and 219 outbreaks in 2018. Currently, only Suzhou (a city in Jiangsu) has implemented two-dose varicella vaccination free of charge, and the others have only given one or no dose. Many studies have shown that one-dose of varicella vaccine was not enough to prevent school outbreaks [16,17]. A two-dose vaccination strategy should be mandatory in China. In addition, an optimal schedule of vaccination should be designed based on many other factors. A modelling study from Italy demonstrated that coverage, efficacy, number of doses or dosing interval are also critical success factors for vaccination [18].

It is important to establish efficient strategies to control the spread of varicella in a highly susceptible population. The major objective of this research is to estimate the optimal strategy for varicella vaccination. First, we discussed the optimal time of second dose inoculation. Second, we analyzed the impact of other parameters on the varicella epidemic, which are vaccination coverage rate, vaccination efficacy and contact rate. Furthermore, simulations were conducted to investigate the effect of the two-dose vaccination coverage rate and related strategies and measures on varicella transmission.

## 2. Methods

### 2.1. Data Source

Every month from 2005 to 2018, we collected the reported varicella cases, either clinically diagnosed or lab-confirmed from Jiangsu Provincial Center for Disease Control and Prevention. All cases were voluntarily reported by local doctors.

### 2.2. Population

According to demographic information from the Statistics Bureau of Jiangsu Province, the population was divided into seven age groups, and k=1,2,3,4,5,6,7 was used to represent the >1 year (0–1), 1 ≤ age < 3 (1–3), 3 ≤ age < 5 (3–5), 5 ≤ age < 10 (5–10), 10 ≤ age < 15 (10–15), 15 ≤ age < 20 (15–20), >20, respectively. The number of new births per year was acquired from the Statistics Bureau [19]. For simplicity, the difference in disease-related mortality was ignored, which was taken as 7‰ from Statistics Bureau in all age groups [19].

According to the primary epidemiological properties of varicella, we divided the total population of each age group at time t into three compartments: susceptible (S(k,t)), infected (I(k,t)) and recovered or immune (R(k,t)). Considering that vaccination in the susceptible may fail, we further divided the susceptible of each age groups into susceptible without vaccination ($S_{0}(k,t)$), the susceptible with first dose vaccination but failed ($S_{1}(k,t)$) and the susceptible with second dose vaccinations but all failed ($S_{2}(k,t)$).

### 2.3. Vaccination Strategy

Children below ten years old (0–1, 1–3, 3–5 and 5–10 age groups) in any condition can be vaccinated with the first or second dose. We assumed that the first dose vaccination can only be given to individuals susceptible but not yet vaccinated ($S_{0}(k,t)$), and the second dose vaccination to those susceptible with failed primary vaccination ($S_{1}(k,t)$). Note that for individuals who were susceptible (due to primary failure) but whose age was less than one year old ($S_{1}(1,t)$) they were not given the second dose, because the guidelines recommend that the minimum interval between the two doses must be greater than 12 months [20].

The effective rates of the first and second doses were presented as constants $\varepsilon_{1}$ and $\varepsilon_{2}$ respectively, and the corresponding vaccination coverage rates in the susceptible people at *k*th age group as $\eta_{1}(k)$ and $\eta_{2}(k)$ respectively. We obtained the vaccination outcomes in each age group (Table 1). Here, $V_{0}^{f}(k,t)$ and $V_{0}^{s}(k,t)$ were defined as individuals with failed and successful first dose vaccination in the *k*th age group, respectively. Similarly, $V_{1}^{f}(k,t)$ and $V_{1}^{s}(k,t)$ were the individuals with failed and successful second dose vaccination in the *k*th age group, respectively.

### 2.4. Model Structure

According to the above mentioned population division and vaccination strategy, as well as the epidemic characteristics of varicella, we set up the compartments in each age group (Figure 1). For the *k*th age group, the flowchart describes how individuals can move among the five mutually exclusive states. Considering the interpersonal behaviors, we roughly assumed that every susceptible person may contact all infected persons and become infected.

Based on the flowchart in the graphical abstract image, the following deterministic SIR dynamic model with age structure can be proposed to mimic the varicella epidemic in Jiangsu province, China.
(1)$$\left\{\begin{array}{l} \frac{dS_{0}(k,t)}{dt}=F_{0}(k-1,t)-V_{0}^{f}(k,t)-V_{0}^{s}(k,t)-\beta (k)S_{0}(k,t)\overset{7}{\underset{k^{\prime }=1}{\sum }}I(k^{\prime },t)-F_{0}(k,t)-dS_{0}(k,t),\\ \frac{dS_{1}(k,t)}{dt}=F_{1}(k-1,t)+V_{0}^{f}(k,t)-V_{1}^{f}(k,t)-V_{1}^{s}(k,t)-\beta (k)S_{1}(k,t)\overset{7}{\underset{k^{\prime }=1}{\sum }}I(k^{\prime },t)-F_{1}(k,t)-dS_{1}(k,t),\\ \frac{dS_{2}(k,t)}{dt}=F_{2}(k-1,t)+V_{1}^{f}(k,t)-\beta (k)S_{2}(k,t)\overset{7}{\underset{k^{\prime }=1}{\sum }}I(k^{\prime },t)-F_{2}(k,t)-dS_{2}(k,t),\\ \frac{dI(k,t)}{dt}=\beta (k)\left(S_{0}(k,t)+S_{1}(k,t)+S_{2}(k,t)\right)\overset{7}{\underset{k^{\prime }=1}{\sum }}I(k^{\prime },t)-cI(k,t)-dI(k,t),\\ \frac{dR(k,t)}{dt}=F_{R}(k-1,t)+V_{0}^{s}(k,t)+V_{1}^{s}(k,t)+cI(k,t)-F_{R}(k,t)-dR(k,t), \end{array}\right.$$

For simplicity, (1) is only a sub-model of the *k*th age group, and the full model of varicella transmission is given in Appendix A. Here $F_{i}(k-1,t)$ represents the transfer rate from (*k* − 1)th age group to the *k*th age group of the susceptible population who have received *i*th vaccination at time t, i=0,1,2 and k=2,3,⋯,7. $F_{R}(k-1,t)$ represents the transfer rate from (*k* − 1)th age group to the *k*th age group of the recovered population.

Since the infectious period of varicella is only one month, we assumed that all the infected people recovered at the same age of infection, so they do not flow between different age groups, and the transfer due to age growth only appeared in the first, second, third and last equations in (1). The value of parameters and initial conditions were detailed in Appendix A.

### 2.5. Simulations

Based on the reported varicella cases in all age groups and the proposed dynamic model, an adaptive Metropolis-Hastings algorithm was used to carry out the Markov Chain Monte Carlo (MCMC) procedure to estimate the undetermined parameters in model (Appendix A for details). The algorithm runs for 2 × 10^6^ iterations with a burn-in of 10^6^ iterations. The Geweke convergence diagnostic method was employed to assess the convergence [21]. The median values of the estimated parameters and their corresponding first and third quartiles (Q1–Q3) were listed in Appendix A, and their distribution were depicted in Appendix A.

### 2.6. Basic Reproduction Number

Basic reproduction number, *R*_0,_ is a key parameter in infectious epidemiology, which indicates the expected number of secondary infections produced by an infected case. If *R*_0_ < 1, this disease cannot invade the population and will eventually disappear, otherwise, if *R*_0_ > 1, the disease will continue to spread and become endemic [22]. See Appendix A for the detailed calculation process.

### 2.7. Global Sensitivity Analysis

Because the basic reproduction number represents the epidemic intensity of the disease, we used it as a dependent variable in Latin Hypercube Sampling (LHS). Partial rank correlation coefficients (PRCCs) were used to examine the global sensitivity of the parameters in the model. Clearly, expanding vaccination coverage, reducing the contact frequency with infected population, and improving vaccination efficacy are all potentially feasible strategies for better preventing and controlling varicella epidemics. The effect of these three parameters on the basic regeneration number is discussed further in Appendix A.

### 2.8. Vaccination Scenarios

In this study, in addition to the effective contact rate, it has been found that increasing the coverage rate of vaccination is more conducive to disease control than increasing the efficiency of vaccination. Thus, in order to simulate the effect of the second vaccination since 2018, we simulated the long-term trend of varicella epidemic in the next 50 years according to the following two scenarios: (1) increase the coverage of the second vaccination while keeping the contact rate unchanged in the model; (2) increase the coverage of the second vaccination when the contact rate decreases in the model, i.e., assuming that the contact rate is 80% of its original baseline value.

## 3. Results

### 3.1. Varicella Data and Underdetermined Parameters

The annual varicella cases in Jiangsu Province from 2005 to 2018 (red circles) and the best-fit simulation of full model with the fitted parameters (Figure 1). According to the goodness-of-fit test for nonlinear fitting [23], our model achieved 0.8211 in the overall goodness, which indicates that the model results were in good agreement with the reported data. We observed that the reported varicella cases in all age groups have been growing exponentially from 2005 to 2018 in Jiangsu Province.

### 3.2. Basic Reproduction Numbers

The overall basic reproduction number can be decomposed as $R_{0}=\overset{7}{\underset{k=1}{\sum }}R_{0}(k)$, where $R_{0}(k)$ was the contribution in each age group (See Appendix A for detailed procedures). From Figure 2A, we observed that the overall basic reproduction number was 1.831 ± 0.078, indicating that varicella has become an endemic disease in Jiangsu Province. Furthermore, from high to low, the contribution of each age group to the epidemic was 5–10-years-old (0.747 ± 0.042, 40.80%), 10–15-years-old (0.401 ± 0.035, 21.90%), 3–5-years-old (0.219 ± 0.012, 11.98%), more than 20-years-old (0.213 ± 0.027, 11.66%), 15–20-years-old (0.128 ± 0.020, 6.98%), 1–3-years-old (0.085 ± 0.005, 4.62%), 0–1-year-old (0.038 ± 0.003, 2.06%).

### 3.3. Sensitivity Analysis

Taking the basic reproduction number as an output variable, we calculated the PRCCs of the three factors, including vaccination coverage rate, vaccination efficacy and effective contact rate. From Figure 2B, we found that almost all parameters were sensitive to the basic reproduction number except $\eta_{1}(1)$ (first dose vaccination coverage rate in the first age group) and β(1) (the effective contact rate in the first age group), which can explain the low percentage of basic reproduction number in the total in the first age group (Figure 2A). For the vaccination coverage related parameters, the simulation results showed that the PRCC values of the second dose vaccination coverage were always greater than those of the first dose, which indicates that the second immunization is critical to control the varicella epidemic. The urgent task is to find ways to improve the second vaccination coverage rate. For effective contact rate parameters, we found that the order of PRCC values of each age group was almost as same as that of their corresponding basic reproduction number (Figure 2). The age groups with the largest PRCC values were 5–10-years-old, 10–15-years-old and 15–20-years-old, respectively, and the possible explanation is that these age groups have long-time social activities, such as attending school [24]. The parameters of both vaccinations have a very significant negative impact on the prevalence of varicella.

### 3.4. Association between Basic Reproduction Number and the Three Prevention Factors

According to the basic reproduction number in Appendix A, we simulated the critical basic reproduction number surface (Figure 3A) and the corresponding contour maps (Figure 3B–D) with these parameters and their different combinations.

### 3.5. Impact of Second Dose Vaccination Coverage Rate on the Varicella Transmission

Based on the methods described earlier, Figure 4 shows the qualitative prediction of a varicella epidemic if we increase second dose vaccination coverage under two scenarios. The results indicated that increasing the coverage rate is always beneficial to control the varicella epidemic, which can not only reduce the basic reproduction number, but also reduce the peak of the epidemic or make the peak earlier. Based on Figure 4, it is not surprising that reducing the contact rate at the same time (Figure 4B) can better control the epidemic than keeping the contact rate unchanged (Figure 4A).

## 4. Discussion

In this study, we developed a dynamic compartmental model to explore the optimal age for second dose varicella vaccination and benefits of increasing the coverage rate. Our model fitted well with annually reported varicella cases from 2005 to 2018 in Jiangsu Province, China. Based on the SIR model, the estimated basic reproduction number was 1.831 ± 0.078, indicating that varicella is endemic in Jiangsu Province. The WHO recommends that in countries where varicella is a major public health issue, varicella vaccination should be introduced into their routine immunization programs [24]. To date, the free second dose of varicella vaccination is still not fully promoted in Jiangsu Province.

To our knowledge, this study is the first to use a mathematic model to evaluate the optimal age for the varicella vaccination strategy in Jiangsu Province. We concluded that the optimal age for second dose varicella vaccination is 5–10 years old, with more evident results illustrated in Figure 2. For the two-dose vaccination schedule there is continuous debate about whether a short-term (months) or longer term (years) interval should be kept. Some hold that a short interval between two doses would lead to a lower risk of varicella breakthrough infections, but with less durable protection over time, and a long interval would produce durable immunity, but children might face a higher risk of breakthrough infections in their early life [25]. A recent study [25] used an agent-based-model to compare the effectives of the two vaccination schedules: (1) first dose at 12 months, second dose at 18 months, and (2) first dose at 12 months, second dose at 4–6 years. In that study, the model predicted that the second dose at 18 months may be more effective than that at 4–6 years, but the difference between the two schedules was insignificant. Therefore, policymakers should consider other factors to determine the appropriate varicella vaccine schedule. Considering factors like security, simplicity and persistence of immunity, the second dose should be given to an older group. Since varicella frequently breaks out in the kindergarten and primary schools [26], it is reasonable to come to this conclusion.

Several other factors also determine the effectiveness of UVV. Sensitivity analysis confirmed a stronger negative correlation between the coverage rate and basic reproduction number, especially for the second dose (Figure 4B). However, model simulation revealed that the second dose coverage rate was far lower than the first dose (about 5% vs. 65%). The WHO has already recommended that vaccines should cover above 80% of the population [27]. High vaccination coverage not only effectively reduces the incidence of breakthrough varicella, but also generates a herd protection effect [28]. We found that when the second dose coverage rate increased by 7.347 times, the varicella transmission could be terminated, at the same time, with contract rate reduced by 80%, and varicella cases could be reduced 3.751 times (Figure 4. Holl et al. conducted a model-based study, suggesting that increasing the coverage rate can compensate for less effective vaccination strategies, such as a longer interval between two doses and lower second dose efficacy [18]. Similarly, the combination of these measures can control varicella more effectively.

Our kinetic model provides an effective method to study the effect of the second dose of vaccine on the spread of the disease. Dynamic models are widely used to generate optimal policies for controlling the spread of infectious diseases [29]. Using mathematical models, we can analyze big data or make predictions based on a limited data in epidemiology.

There are several limitations in our study. First, we only considered the varicella cases from Jiangsu Province, thus the generalization of the conclusion was limited to some extent. Secondly, this paper only studied the effect of vaccination strategies on varicella, ignoring the effect on herpes zoster. This is because the study was not designed to prove the causality between herpes zoster and varicella vaccination. Third, many factors affecting the implementation of vaccination strategy should be taken into consideration, including duration of vaccination effectiveness, catch-up program, breakthrough varicella, and financial cost.

## 5. Conclusions

Increasing vaccine coverage can improve the prevention and control of varicella. The two-dose varicella vaccination should be compulsory, and the optimal age of two-dose vaccination is 5–10 years in Jiangsu Province, China.

## Figures and Tables

**Figure 1 tropicalmed-08-00017-f001:**
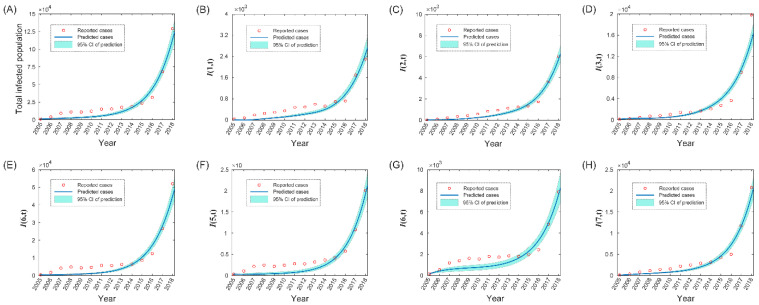
Illustration of the fitting result of the full model from 2005 to 2018 underestimated parameters. The blue solid line and azure shade represent the model-predicted cases and 95% confidence interval (CI). Red circles denote survey-reported cases. (**A**) is the fitting result of total infected population, and (**B**–**H**) is the fitting result of each age group.

**Figure 2 tropicalmed-08-00017-f002:**
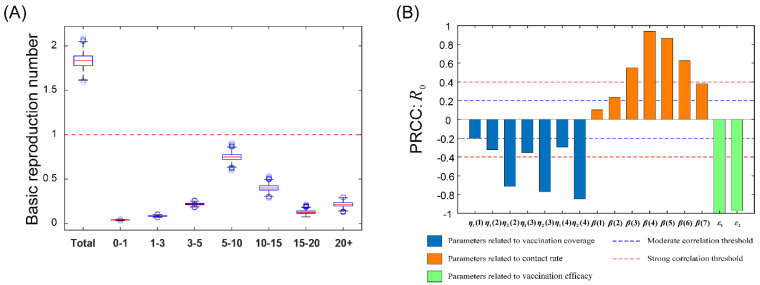
(**A**) Estimated basic reproduction number based on the full model. (**B**) Partial rank correlation coefficients illustrating the sensitivity of basic reproduction number on three types of parameters, named as vaccination coverage rate, vaccination efficacy and effective contact rate, which are shown by blue, green and orange bars, respectively. The red dashed line in (**A**) denotes the critical threshold. The blue and red lines in (**B**) represent moderate and strong correlation thresholds of PRCC values, respectively.

**Figure 3 tropicalmed-08-00017-f003:**
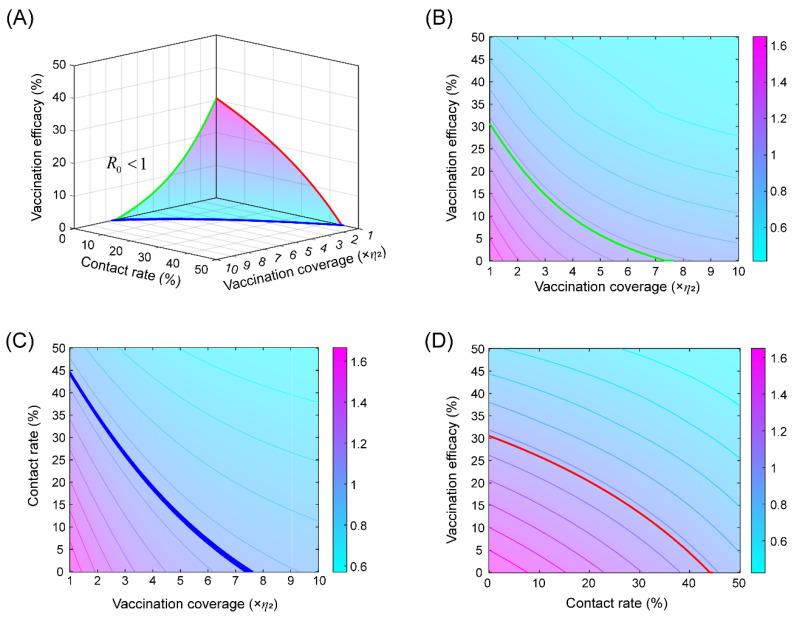
(**A**) The surface of *R*_0_ = 1 in terms of parameters related to vaccination coverage rate, vaccination efficacy and contact rate. The surface above the plane represents *R*_0_ < 1 and otherwise *R*_0_ > 1. The contours of *R*_0_ versus vaccination coverage rate and vaccination efficacy (**B**), versus vaccination coverage rate and contact rate (**C**), versus the contact rate and vaccination efficacy (**D**).

**Figure 4 tropicalmed-08-00017-f004:**
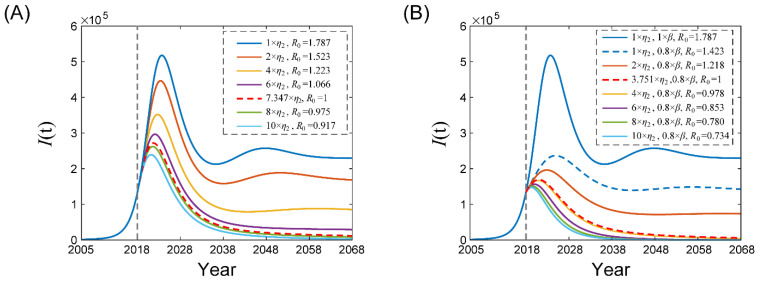
The impact of increasing second dose vaccination coverage rate staring in 2018 on the varicella transmission with (**A**) unchanged contact rate and (**B**) reduced contact rate (80% baseline). *η*_2_ represents all parameters related to second dose vaccination. *β* represents all parameters of contact rate.

**Table 1 tropicalmed-08-00017-t001:** Vaccination outcomes in each age group.

Age Group	$V_{0}^{f}(k,t)$	$V_{0}^{s}(k,t)$	$V_{1}^{f}(k,t)$	$V_{1}^{s}(k,t)$
0–1 year	$\eta_{1}(1)(1-\varepsilon_{1})S_{0}(1,t)$	$\eta_{1}(1)\varepsilon_{1}S_{0}(1,t)$	NA	NA
1–3 years	$\eta_{1}(2)(1-\varepsilon_{1})S_{0}(2,t)$	$\eta_{1}(2)\varepsilon_{1}S_{0}(2,t)$	$\eta_{2}(2)(1-\varepsilon_{2})S_{1}(2,t)$	$\eta_{2}(2)\varepsilon_{2}S_{1}(2,t)$
3–5 years	$\eta_{1}(3)(1-\varepsilon_{1})S_{0}(3,t)$	$\eta_{1}(3)\varepsilon_{1}S_{0}(3,t)$	$\eta_{2}(3)(1-\varepsilon_{2})S_{1}(3,t)$	$\eta_{2}(3)\varepsilon_{2}S_{1}(3,t)$
5–10 years	$\eta_{1}(4)(1-\varepsilon_{1})S_{0}(4,t)$	$\eta_{1}(4)\varepsilon_{1}S_{0}(4,t)$	$\eta_{2}(4)(1-\varepsilon_{2})S_{1}(4,t)$	$\eta_{2}(4)\varepsilon_{2}S_{1}(4,t)$

## Data Availability

All data generated or analysed during this study are included in this published article. The data that support the findings of this study are available from the corresponding author upon reasonable request.

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
