# Peer review of "A Dynamic Compartmental Model to Explore the Optimal Strategy of Varicella Vaccination: An Epidemiological Study in Jiangsu Province, China"

_tropicalmed, 2022, doi:10.3390/tropicalmed8010017_

Round 1

Reviewer 1 Report

General comment: Here, the authors present a dynamic mathematical model to investigate the optimal varicella vaccination strategy in Jiangsu Province, China. Universal vaccination against varicella is not mandatory in China and vaccination scheme is not defined. Based on the results of this model, the authors propose an age of 5-10 years as optimal for the second dose of varicella vaccination. They also found that increasing vaccination coverage and reducing contact rates would be most beneficial for controlling varicella epidemics.The findings are interesting, novel and could be of interest for defining optimal vaccine strategy in China.

However,  my impression is that the manuscript would be a bit confusing for readers. Shortening as well as simplification is needed. The following suggestions could be used to improve the manuscript:

1) Introduction

Lines 39-50: It is necessary to shorten the introduction, especially the general part about the virus/disease which are well known to most readers. Instead, the authors could show the  age-specific incidence rates of varicella  in the previous ten-year period  derived from surveillance data ( for the Province). Also, they could specify recommendations for vaccination against varicella i.e. to whom varicella vaccine is currently recommended ( for example seronegative healthy children, adolescents, adults, specific vulnerable groups etc)?

2) Methods...generally too long and unclear- I advise shortening and simplification as much as possible. Specifically:

Data source:

Line 77: Varicella surveillance needs to be better explained. Although you wrote that reporting is voluntary, please explain whether there is sentinel surveillance, is it case-based  and how up-to-date the reporting is, i.e. how much it reflects the actual epidemiological situation in the province/country?

Population:

Line 81: The division of age groups is shown in such a way that the age groups overlap ( 0-1,1-3,3-5,5-10...). That is not allowed. Please make necessary corrections. 

Line 83:  I am not sure that difference in disease-relate mortality could be ignored, and why you choose the same rate  ( 7%..) as approximate mortality for all age groups ?

Vaccination strategy

Line 95: Although this is fictive mathematical model it should reflect real-life scenario, as faithfully as possible. Please explain why and how you defined the group "susceptibles with failed  primary vaccination". Does this apply to single-dose vaccinees who are sero-negative ( a lack of IgG varicella specific antibodies in serum)  after a certain time since vaccination? How do you define those with successful first dose vaccination-are they those who are  sero-positive  after a certain time after vaccination? This question also refers to the individuals with failed and successful second dose vaccination ( lines 105-106).

Global sensitivity analysis

Lines 154-174. Too extensive. Needs shortening and simplification. I advise you to simplify the text as much as possible or to transfer it to the supplementary material.

Vaccination scenario

Lines 176-180: Difficult to understand and unclear. Please simplify and explain better. Line 178: confusing to read.

Results.

Generally, in many places the author comments on the results (for example lines 211, 216, 240-244). The results must be presented concisely, and the comments of the results should be transferred to the discussion.

Discussion

Lines 252-259- this part of paragraph should be moved and merged with the text ( lines 59-65) in the Introduction section- it refers to the same topic.

Lines:253-254- repeating ( it was already mentioned in the introduction-line 59). Please avoid repetition in the manuscript.

Please answer: if any VZV sero-surveys have been done in China recently? If yes, explain how do the results presented in the manuscript fit with the VZV seroprevalence data generated in these studies?

Be more practical and precise with respect to the recommended varicella vaccination schedule arising from the results of this study. For example, would vaccinating children before entering elementary school be the best option for a second dose (especially if other vaccines are given at that time)? Based on your results, what would be the recommendations for policy makers in your Province or China? Do you have a suggestion for additional studies that would build on your results?

Author Response

Dear Editor,

We wish to thank you for your thoughtful review of our manuscript Ref. No.: tropicalmed-2060727 entitled " A dynamic compartmental model to explore the optimal strategy of varicella vaccination: an epidemiological study in Jiangsu Province, China".

We have carefully considered each of the comments, and an itemized response to each of the critiques as follows:

Reviewer #1:

Question1:

1) Introduction

Lines 39-50: It is necessary to shorten the introduction, especially the general part about the virus/disease which are well known to most readers. Instead, the authors could show the age-specific incidence rates of varicella in the previous ten-year period derived from surveillance data (for the Province).

Answer1:

Thank you very much for your suggestion. We have been simplifying the section on virus/disease in lines 39-50. As introduction has already described the varicella epidemic and vaccination in Jiangsu in 2017-2018, this paragraph only further describes the risk of varicella infection in children aged 4-6 years in Jiangsu.

Question2:

Also, they could specify recommendations for vaccination against varicella i.e. to whom varicella vaccine is currently recommended (for example seronegative healthy children, adolescents, adults, specific vulnerable groups etc)?

Answer2:

Thank you very much for your suggestion. In the results section of this study, we found the highest contribution to the varicella epidemic among the various age groups (0.747 ± 0.042, 40.80%) for the 5-10 years age group. The PRCC values were also highest in the 5-10 years age group when sensitivity analyses were conducted. This may be due to the fact that this age group has a longer period of social activity, such as going to school [1]. Therefore, priority needs to be given to completing two doses of vaccine for children in this age group.

Question3:

Methods...generally too long and unclear- I advise shortening and simplification as much as possible. Specifically:

Data source:

Line 77: Varicella surveillance needs to be better explained. Although you wrote that reporting is voluntary, please explain whether there is sentinel surveillance, is it case-based  and how up-to-date the reporting is, i.e. how much it reflects the actual epidemiological situation in the province/country?

Answer3:

Thank you very much for your suggestion. All data in this study were obtained from data summarized by the Center for Disease Control and Prevention of Jiangsu Province. According to the requirements of the Jiangsu Provincial Health and Wellness Commission, chickenpox is managed according to category C infectious diseases. Each medical institution should report information on chickenpox cases through the Chinese Information System for Disease Prevention and Control (CISDCP) within 24 hours. Therefore, this data can truly reflect the epidemiological characteristics of chickenpox in Jiangsu Province.

Question4:

Population:

Line 81: The division of age groups is shown in such a way that the age groups overlap ( 0-1,1-3,3-5,5-10...). That is not allowed. Please make necessary corrections.

Answer4:

Thank you very much for your suggestion. Age groups are divided into: age <1 years old, 1 ≤ age <3, 3 ≤ age <5, 5 ≤ age <10, and we have made changes to the age groupings in all corresponding places throughout the text.

Question5:

Line 83:  I am not sure that difference in disease-relate mortality could be ignored, and why you choose the same rate ( 7%..) as approximate mortality for all age groups ?

Answer5:

Thank you very much for your advice. Notice that varicella is a self-limited disease, and it is a relatively mildly symptomatic infectious disease. Although patients with severe varicella attacks or poor underlying status may have serious complications, which may even lead to death, these complications are usually very rare. Especially, there are no clinical statistics available on deaths diagnosed mainly from varicella and therefore no clear reference range can be derived on the disease related mortality. Therefore, considering the fact that its mortality is usually very low, in order to reduce the number of undetermined parameters, we ignore the disease related mortality and only consider the natural mortality of all age groups in model.

Question6:

Vaccination strategy

Line 95: Although this is fictive mathematical model it should reflect real-life scenario, as faithfully as possible. Please explain why and how you defined the group "susceptibles with failed primary vaccination". Does this apply to single-dose vaccinees who are sero-negative (a lack of IgG varicella specific antibodies in serum) after a certain time since vaccination? How do you define those with successful first dose vaccination-are they those who are sero-positive after a certain time after vaccination? This question also refers to the individuals with failed and successful second dose vaccination (lines 105-106).

Answer6:

Thank you very much for your advice. "susceptibles with failed primary vaccination" is defined as primary vaccination failure when there is no protective immune response after a single dose of varicella vaccine. A patient with a successful first dose is defined as having IgG varicella-specific antibodies in the serum after vaccination. This definition also applies to people who have failed and succeeded in the second dose of vaccination.

IgG-specific antibodies are used to distinguish between successful varicella vaccination because: IgG positivity for varicella-zoster virus indicates previous varicella infection or varicella vaccination. IgG antibodies are protective antibodies that indicate no future reinfection. According to the Hangzhou 2019 cross-sectional study, the effectiveness of one dose of varicella vaccine was 91% and the effectiveness of two doses of varicella vaccine was 98%.

Question7:

Global sensitivity analysis

Lines 154-174. Too extensive. Needs shortening and simplification. I advise you to simplify the text as much as possible or to transfer it to the supplementary material.

Answer7:

Thank you very much for your suggestion, we have streamlined the paragraph and added at the end of the paragraph “The effect of these three parameters on the basic regeneration number is discussed further in Appendix E.”

Question8:

Vaccination scenario

Lines 176-180: Difficult to understand and unclear. Please simplify and explain better. Line 178: confusing to read.

Answer8:

Many thanks for your comment. We are very sorry that the description is not clear enough to cause you confusion. In the revised version (lines 162-169), we have rewritten the subsection as follows:

Vaccination scenarios

In this study, besides the effective contact rate, it has been found that increasing the coverage rate of vaccination is more conducive to disease control than increasing the efficiency of vaccination. Thus, in order to simulate the effect of the second vaccination since 2018, we simulated the long-term trend of varicella epidemic in the next 50 years according to the following two scenarios: (1) increase the coverage of the second vaccination while keeping the contact rate unchanged in the model; (2) increase the coverage of the second vaccination when the contact rate decreases in the model, i.e., assuming that the contact rate is 80% of its original baseline value. -

Question9:

Discussion

Lines 252-259- this part of paragraph should be moved and merged with the text ( lines 59-65) in the Introduction section- it refers to the same topic.

Answer9:

Thank you very much for your suggestion. The discussion duplicates the description in the introduction of the policy of free second dose varicella vaccination in Suzhou. As we know, the public health policy on varicella vaccination is an important factor in varicella vaccination rates. We have merged the two descriptions of the second dose of varicella vaccine into the introduction. Delete in Discussion “However, free varicella vaccination is not yet universal in Jiangsu Province. Currently, 2-dose varicella vaccination is only implemented in Suzhou free of charge. The most common 2-dose vaccination schedule comprises a first dose at 12-18 months and a second dose at 4-6 years old. Alternatively, the second dose can be given to children below 4 years old, with an interval of 3 months or more between two doses [2]. At present, some provinces and cities in China have started to implement the 2-dose vaccination strategy, such as Beijing, Tianjin, Shanghai and Guangdong, which also use the above vaccination strategy.”, and added “To date, the free second dose of varicella vaccination has still not been fully promoted in Jiangsu Province.”

Question10:

Please answer: if any VZV sero-surveys have been done in China recently? If yes, explain how do the results presented in the manuscript fit with the VZV seroprevalence data generated in these studies?

Answer10:

Thank you very much for your suggestion. There are few studies on VZV serological investigations in China and we have searched various serological studies in China and abroad to estimate the efficiency of antibody vaccination. Results from a study of 3140 children in Suzhou showed that the anti-varicella virus IgG seropositivity rate of 54.4% in children vaccinated against varicella was significantly higher than that of unvaccinated children (49.2%). (χ2 = 8.206, P = 0.004). Among vaccinated children, the detection rate of varicella IgG antibodies increased with age, with 49.4%, 50.9% and 58.9% detected in the 4-, 5- and 6-year old groups, respectively [3]. Another study of healthy children aged 1-9 years who received live attenuated varicella vaccine throughout Jiangsu showed that only 43.1% of healthy children in Jiangsu had varicella vaccine coverage in 2016, with only 57.1% seropositive after 1 dose of vaccination [4]. In a previous Korean study, anti-VZV IgG seropositivity was 49% at 4 years of age, 62% at 5 years of age and 70% at 6 years of age, respectively [5]. Another meta-analysis assessing the effectiveness of varicella vaccination worldwide showed that the effectiveness against varicella after a single dose of varicella vaccine ranged from 76% to 85% [6]. This study assumed [Appendix B] a 60% effectiveness rate after the first dose of vaccine and a 75% effectiveness rate after the second dose of vaccine.

Question11:

Be more practical and precise with respect to the recommended varicella vaccination schedule arising from the results of this study.

For example, would vaccinating children before entering elementary school be the best option for a second dose (especially if other vaccines are given at that time)?

Based on your results, what would be the recommendations for policy makers in your Province or China? Do you have a suggestion for additional studies that would build on your results?

Answer11:

Thank you very much for your advice. For the recommendation of the first dose, we believe that vaccination at 12 months is more appropriate. Although other models predict that a second dose at month 18 is more effective [7], the difference between these two regimens is not significant compared to a second dose at 4-6 years of age. In particular, children in the 5-10 year age group have increased social activity and a significantly higher incidence of varicella. A second dose of vaccine at 18 months results in less durable protection and an increased risk of breakthrough infection at 5-10 years [7]. Therefore, taking into account several characteristics such as safety, simplicity and durability of immune protection, this study suggests that policy makers would be more effective in vaccinating children with a second dose at 4-6 years of age.

Reviewer 2 Report

Varicella or chickenpox is a highly contagious disease primarily in children, which spreads by the inhalation of aerosols or direct contact.  The virus usually causes only relatively mild disease in children, but it can also remain latent in sensory nerve ganglia for decades before causing painful herpes zoster later in life.  In the US and several other countries, a two-dose vaccination protocol has been in place for several years and the disease is well-controlled by it.  However, this is not the case in China and, in Jiangsu Province, the number of varicella infections has progressed exponentially since 2014. 

In this excellent theoretical manuscript, the authors develop a dynamic model to examine the efficacy of various vaccination strategies and schedules on the incidence of varicella in this province with the goal of identifying the optimal strategy for prevention of varicella.  Initially establishing that a two-dose regimen is vital, the group analyzed the optimal scheduling of the second vaccine dose.  Their findings argue strongly that the optimal schedule for vaccination is a primary dose at 12-18 months and a second dose at 5-10 years of age.

This is considered a thoughtful and thorough analysis of varicella vaccination protocols and their efficacy.  While the importance of a two-dose varicella vaccine regimen is evident by the success of the protocol in several countries, the data presented here certainly verify that finding and further establish the optimal scheduling for the second does in particular.  The authors are to be commended for their identification of the limitations to their study, i.e., that it does not address the effect on herpes zoster and considers only the situation in a single Chinese province, although one strongly suspects that the latter is only a minor limitation, as it is likely that these findings would apply to any population. 

The manuscript is considered acceptable for publication once a few very minor points are addressed:

1)    Are the exact same vaccines used throughout the world?

2)    In the interest of completeness, it would be helpful if the authors at some point provided basic information concerning the nature of the vaccines used (inactivated, subunit, etc.). 

Author Response

Reviewer #2:

Question1:

Are the exact same vaccines used throughout the world?

Answer1:

All live lyophilized attenuated varicella vaccine is currently used in Jiangsu Province. Live lyophilized varicella attenuated vaccine is currently the main preventive vaccine for varicella used worldwide. The recombinant herpes zoster vaccine is a non-live, recombinant subunit vaccine that, in combination with VZV and an adjuvant system, enhances the body’s specific cellular and humoral immune response to VZV, thereby preventing the development of herpes zoster. The main target population is people aged 50 years and older, a group with generally reduced resistance, where the addition of adjuvants can improve the immunogenicity of the antigen and the effectiveness of the vaccine, playing a key role in the protective efficacy of the vaccine and enhancing and modulating the immune response. It is currently only approved in Canada, the USA, the EU, Japan and China. Research on varicella vaccines for children is still focused on lyophilized live attenuated varicella vaccines.

Question2:

In the interest of completeness, it would be helpful if the authors at some point provided basic information concerning the nature of the vaccines used (inactivated, subunit, etc.).

Answer2:

The varicella vaccines used in Jiangsu are lyophilized live attenuated varicella vaccines. There are five Jiangsu suppliers of varicella vaccine, namely Changchun Qijian, Shanghai Institute, Changchun Encyclopedia, Changchun Changsheng and Dalian Kexing.

Round 2

Reviewer 1 Report

The authors answered sufficiently to all the questions and comments raised in the review. The manuscript has been significantly improved and deserves to be published.